# Dual-Cure Adhesives Using a Newly Synthesized Itaconic Acid-Based Epoxy Acrylate Oligomer

**DOI:** 10.3390/polym15153304

**Published:** 2023-08-04

**Authors:** Hae-Chan Kim, Yong-Rok Kwon, Jung-Soo Kim, Ju-Hee So, Dong-Hyun Kim

**Affiliations:** 1Material & Component Convergence R&D Department, Korea Institute of Industrial Technology (KITECH), Ansan-si 15588, Republic of Korea; coolskawk@kitech.re.kr (H.-C.K.); yongrok@kitech.re.kr (Y.-R.K.); kimjungsoo11@kitech.re.kr (J.-S.K.); jso@kitech.re.kr (J.-H.S.); 2Department of Materials Science and Chemical Engineering, Hanyang University, Ansan-si 15588, Republic of Korea

**Keywords:** itaconic acid, epoxy acrylate oligomer, dual-cure

## Abstract

Herein, a novel biomass-derived itaconic acid (IA)-based epoxy acrylate oligomer (EAO) is synthesized by means of the esterification reaction of the epoxy group of bisphenol A diglycidyl ether (BADGE) with the carboxylic group of IA. The detailed chemical structure of the as-prepared bisphenol A diglycidyl ether diitaconate (BI) is characterized via the KOH value, FT-IR spectrum, and ^1^H-NMR spectrum. Further, a dual-cure adhesive system is formulated using BADGE, acrylic acid, and trimethylolpropane triacrylate with various BI contents, and the adhesive performance is investigated by measuring the thermal stability, adhesive properties, pencil hardness, and surface energy properties. Thus, the dual-cure adhesive with a BI content of 0.3 mol is shown to provide excellent thermal stability, along with an adhesive strength of 10.7 MPa, a pencil hardness of 2H, and a similar surface energy to that of a typical polycarbonate film. In addition, the properties of the BI-based dual-cure adhesive are compared with those of the dual-cure adhesives based on bisphenol A glycerolate diacrylate or bisphenol A glycerolate dimethacrylate.

## 1. Introduction

With increasing efforts towards carbon neutrality and growing concerns about fossil fuel depletion, the plastics industry has been rapidly shifting from traditionally used petroleum-based polymer materials to more sustainable alternatives. Among these alternatives are bio-based thermoplastic resins derived from sources like cellulose, starch, polylactic acid (PLA), and polyhydroxyalkanoate (PHA) [1,2,3,4,5]. However, the development of bio-based thermosetting resins has proven challenging due to their comparatively poor physical properties [6,7]. Despite these challenges, there has been a push towards finding viable bio-based alternatives in this domain.

One such promising biomass-derived material is itaconic acid (IA) [8], which has gained attention for its potential as a natural and sustainable alternative. IA is industrially produced through the fermentation of carbohydrates like lactic acid and starch or via the combustion of natural citric acid [9]. Notably, the IA molecule contains two carboxyl groups and one vinyl group, which forms part of a vinyl acid moiety similar to that found in acrylic acid (AA). Acrylic compounds, including AA, generally possess linear structures and lower thermodynamic stabilities. To address this, monomers and crosslinking agents are used to create thermosetting resin networks with enhanced adhesion, reactivity, chemical resistance, hardness, and scratch resistance [10,11].

Dual curing, also known as co-curing, is a process that combines two curing mechanisms—typically UV- or light-induced curing followed by thermal curing. This approach provides enhanced curing control, improved material properties, faster production speed, and reduced energy consumption [12]. Examples of dual-cure adhesives (DCAs) include epoxy acrylate oligomers (EAOs), dual-cure silicone adhesives [13], and dual-cure polyurethane adhesives [14]. These adhesives offer versatility and performance benefits, making them suitable for various manufacturing processes requiring fast curing and reliable bonding.

This study aims to evaluate the potential of IA-based EAOs for partially replacing EAOs solely derived from petroleum products. To provide a meaningful comparison, the properties of the as-prepared IA-based EAO are also compared with those of commonly used dual-cure adhesives based on bisphenol A glycerolate diacrylate or bisphenol A glycerolate dimethacrylate. By introducing and assessing the properties of this bio-based EAO, the authors seek to contribute important information to the ongoing development of sustainable and eco-friendly alternatives in the field of thermosetting resins.

In summary, this research focuses on the synthesis and characterization of a novel bio-based EAO derived from itaconic acid (IA), exploring its potential as an eco-friendly substitute for petroleum-based epoxy resins. By addressing the increasing demand for sustainable materials and renewable resources, this study represents a significant step towards advancing green chemistry and fostering more environmentally conscious industrial practices.

## 2. Materials and Methods

### 2.1. Materials

For the EAO synthesis, IA (99%, Junsei Chemical, Tokyo, Japan) and bisphenol A diglycidyl ether (BADGE; Sigma Aldrich, St. Louis, MO, USA) were used as monomers, Benzyltriethylammonium chloride (BTEAC; 99%, Sigma Aldrich, USA) was used as a catalyst, 1-propanol (99%, Samchun Chemicals, Seoul, Republic of Korea) was used as a solvent, and hydroquinone (≥99%, Sigma Aldrich, St. Louis, MO, USA) was used as an inhibitor. Acrylic acid (AA; 99%, Sigma Aldrich, St. Louis, MO, USA) and trimethylolpropane triacrylate (TMPTA; Sigma Aldrich, St. Louis, MO, USA) were used as a co-monomer and crosslinker, respectively, without purification. For comparison with the synthesized EA oligomer, bisphenol A glycerolate diacrylate (BA; Sigma Aldrich, St. Louis, MO, USA) and bisphenol A glycerolate dimethacrylate (BMA; Sigma Aldrich, St. Louis, MO, USA) were used. The photoinitiator 2,2-dimethoxy-2-phenylacetophenone (DMPA; Sigma Aldrich, St. Louis, MO, USA) and the thermal curing agent 1-methylimidazole (1-MI; ≥99%, Sigma Aldrich, St. Louis, MO, USA) were used without purification. The polycarbonate (PC) film (1T, Hwa-in Science, Seoul, Republic of Korea) was used as purchased.

### 2.2. Preparation of Bisphenol A Diglycidyl Ether Diitaconate (BI)

Bisphenol A diglycidyl ether diitaconate (BI), in which both ends of BADGE were modified, was prepared at a 2:1 equivalent ratio of IA to BADGE (a 1:1 ratio with respect to the epoxy group). The reactants and reaction conditions are shown schematically in Figure 1.

The reactions were performed in a 500 mL four-necked flask equipped with a mechanical stirrer, a reflux condenser, a thermometer, and a needle for the injection of nitrogen gas. For each reaction, IA (0.4 mol) and BADGE (0.2 mol) were stirred in 1-propanol (20 mL) at 250 rpm. After the monomers were mixed vigorously at 60 °C for 10 min, the temperature was gradually increased to 90 °C, and the initial KOH value was measured. The temperature was then maintained at 90 °C, and the KOH was measured at 1 h intervals until no further change was observed. The reaction was then terminated by the addition of BTEAC (1 wt.%) and hydroquinone (0.1 g). The reaction product was allowed to cool and maintained at room temperature for 1 h before washing three times with deionized water to remove any unreacted monomers. Finally, a transparent EAO with high viscosity was obtained.

### 2.3. Preparation of the Dual-Cure Adhesive

The predetermined amounts of AA, TMPTA, BI, and BADGE were stirred together until a homogeneous mixture was obtained. After that, DMPA (0.1 phr) and 1-MI (0.25 phr) were added, followed by stirring and degassing via sonication. The gas-free mixture was then UV-cured at a wavelength of 360 nm with a dose of 300 mJ/cm^2^ for several seconds, followed by thermal curing at 90 °C in a convection oven for 30 min. The compositions and designations of the various dual-cure adhesive (DCA) samples are listed in Table 1. All samples were cured under the same conditions. To minimize chaotic effects, the EAO network components were kept simple and consistent, i.e., no fillers or reinforcing agents were added.

### 2.4. Acid Value

The acid (or KOH) value is the number of milligrams of potassium hydroxide (KOH) necessary to neutralize the fatty acids in 1 g of the sample. This was determined by means of the addition of methanol (100 mL) and a certain weight of the mixture to the sample container of a potentiometric titrator (888 Titrando; Metrohm, Switzerland) and using the tiamo software.

### 2.5. Characterization of the BI

The structure of the BI (in the form of a potassium bromide pellet) was examined via Fourier transform infrared (FT-IR) spectroscopy (NEXUS Instruments; Thermo Nicolet NEXUS 670) (Thermo Fisher Scientific, Waltham, MA, USA) in the scan range of 600–4000 cm^−1^ at a resolution of 1 cm^−1^ and nuclear magnetic resonance (^1^H-NMR) spectroscopy (Varian; Unity Inova 500 MHz) (Agilent, CA, USA).

### 2.6. Differential Scanning Calorimetry (DSC)

The thermal behaviors of the samples were investigated on a differential scanning calorimeter (DSC; Q100, TA Instruments, New Castle, DE, USA) during two cycles of heating and cooling under a N_2_ atmosphere in the temperature range of 30–250 °C, with a heating rate of 10 °C/min.

### 2.7. Thermogravimetric Analysis (TGA)

The thermal stability is one of the most important properties of thermosetting resins. Hence, the thermal stabilities and decomposition behaviors of the samples were evaluated via the commonly used method of thermogravimetric analysis (TGA), which measures the weight loss during heating [15]. The TGA measurements were performed under a N_2_ atmosphere in the temperature range of 30–800 °C at a heating rate of 20 °C/min, and the sample weight loss was plotted as a function of temperature.

### 2.8. Gel Fraction

The gel fractions of the DCAs (5 g) were measured after UV/thermal curing. The cured DCAs were determined by soaking in methyl ethyl ketone (500 mL) for 24 h. The insoluble part was removed by filtration and dried at 60 °C to a constant weight. The gel fraction was then calculated using Equation (1):(1)$$Gel\;fraction=\frac{\omega_{e}}{\omega_{i}}$$
where *ω*_i_ and *ω*_e_ are the weights of the initial and extracted dried DCA, respectively.

### 2.9. Adhesive Strength

The lap shear strength is commonly used to measure adhesive strength. Herein, the lap shear strengths of the dual-cure specimens were tested according to the ASTM D 1002 standard using universal testing equipment (AllroundLine Fmax 10 kN UTM; ZwickRoell, Ulm, Germany), as shown schematically in Figure 2. For this procedure, specimens with adhesion areas of 1 in × 10 mm and an average thicknesses of 40 µm were prepared on PC films.

### 2.10. Pencil Hardness

The pencil hardness measurement is commonly used in industry to evaluate the surface properties of a coating, although the hardness values obtained are more approximate than those obtained via micro-hardness tests [16]. Herein, the pencil hardness was measured based on the ASTM D3363 test method using a pencil hardness tester (HT-6510P; LANDTEK, Guangzhou, China) with a 1000 g loading. For this measurement, the pencil (6B–6H, ChungHwa, Shanghai, China) was held on the device body and maintained at a 45° angle to the test surface.

### 2.11. Surface Energy

The surface energy measurement is important for confirming the compatibility with other materials. Herein, the surface energy was measured using a contact angle analyzer (Phoenix-MT(T); Surface Electro Optics Co., Suwon-si, Republic of Korea) via the sessile drop method. For each sample, the contact angle was measured five times at room temperature, and the surface energy was calculated via analysis of the contact angles of water and ethylene glycol using the Surfaceware 9.0 software.

## 3. Results and Discussion

### 3.1. Characterization of the BI

At the beginning of the reaction shown in Figure 1, the KOH value was 1.69 mg KOH/g. After 10 h, the KOH value had decreased to 0.93 mg KOH/g, thereby indicating 45% conversion. Given that IA has two carboxyl groups per mole, the corresponding reaction yield is 90%. In addition, the number average molecular weight (M_n_) and weight average molecular weight (M_w_) by GPC measurement of BI were 5146 g and 13,243 g, respectively.

The FT-IR spectra of the IA, BADGE, and BI are presented in Figure 3. Here, the main characteristic peaks of the BI (blue line) are the broad absorption band at 3060−3350 cm^−1^ due to the stretching vibrations of the hydroxyl (O−H) group, which was generated by the epoxy ring-opening reaction, and the peak at 930 cm^−1^ due to the reduction of the epoxy group. In addition, the peaks at 911 cm^−1^ and 1638 cm^−1^ due to the vinyl carbon–carbon double bond (C=CH_2_) of the itaconate (black line) [17] are observed in the BI (blue line). Moreover, the absorption peak at 930 cm^−1^ in the BADGE spectum (red line) indicates the presence of the epoxy groups [18], and is not observed in the BI.

The ^1^H-NMR spectrum of the BI is presented in Figure 4. Here, the characteristic peaks at 4.30 and 4.32 ppm indicate the proton in the bond formed by the epoxy ring-opening reaction of the BADGE with the carboxyl group of IA. This is further confirmed by the absence of peaks at 2.50 and 3.04 ppm, whose presence would otherwise indicate the protons in the ethylene oxide ring of unreacted BADGE. Further, the multiple peaks at 5.72 and 6.37 ppm are due to the protons attached to the unsaturated double bond [19]. These results adequately demonstrate the esterification reaction between IA and BADGE.

### 3.2. Curing Characteristics

The non-isothermal (dynamic) curing characteristics of the BI/AA and BI/BADGE mixtures (0.1 mol, respectively) are as shown in Figure 5 to confirm the curing behavior of BI. Thus, in the absence of BI/AA (black profile), the heat flow of the sample exhibits a single exothermic peak at around 140 °C due to the polymerization and crosslinking reaction between the carbon–carbon double bond of BI and AA. By contrast, the presence of an epoxide group results in multiple exothermic peaks during the heating of the BI/BADGE (blue profile). Here, the initial low-temperature peak is due to the radical copolymerization of AA and BI with the double bond of BI, while the high-temperature peak at around 200 °C corresponds to the epoxy ring-opening reaction with the carboxylic acid of BI. These results clearly demonstrate the successful dual reaction point and curing process of the BI systems under non-isothermal curing conditions.

The FT-IR before and after curing of the DCA containing BI are shown in Figure 6. The most characteristic peaks through curing reactions are due to the presence of C=C participating in the photo-cure reaction and the presence of OH peaks of the carboxyl group and epoxy participating in the thermosetting reaction. After performing the curing reaction, the FT-IR confirmed that the three main peaks were not present. Therefore, the photo/thermal curing reaction of DCA was successful under the set curing conditions.

A schematic diagram of the suggested network configuration obtained via the dual (UV/thermal) curing of the BI-containing adhesive system is presented in Figure 7. Here, UV initiation of the reaction between BI, AA, and TMPTA is followed by thermal initiation of the reaction between BI and BADGE. Reaction mechanism involves the formation of a network via UV-induced decomposition of the photoinitiator DMPA. This leads to the initiation through the generation of radicals, facilitating chain growth between acrylic monomers and the carbon double bond (C=C) at the end of the BI backbone. Next, the formation of the network via thermal curing is as follows. The nucleophilic addition of imidazole to epoxy groups is catalyzed by proton donors in a manner similar to the nucleophilic addition to general epoxy groups. Thereafter, an epoxy ring opening reaction and an ester reaction with a carboxyl group at the end of BI are performed.

### 3.3. Gel Fraction

Observing gel fractions represents a convenient method for measuring insoluble fractions, such as those of bridging or network polymers. The gel fractions of the DCA thermally cured at 90 °C for 30 min after UV curing are shown Figure 8. All DCA sample groups containing BI showed a gel fraction of 95% or more. The gel fraction gradually increased with the increase in BI content, due to the increased concentration of C=C bonds in EAOs. In addition, the gel fraction increased due to the crosslinking reaction between the unreacted monomers, unreacted epoxy resin, and the partially acrylated BI. Therefore, the addition of BI to DCA improves the effectiveness of the gel fraction and demonstrates the crosslinking structure with the trifunctional monomer and epoxy resin. In addition, it suggests the presence of unreacted molecules, and the increase in BI content shows higher cross-linking participation through the reduction in the number of unreacted molecules.

### 3.4. Adhesive Strength

The lap-shear adhesion strengths of the DCA_BI_0_ (none), DCA_BI_1_, DCA_BI_2_, and DCA_BI_3_ samples are compared with those of the BA- and BMA-containing samples that were prepared under the same curing conditions in Figure 9. Here, the DCA_BI_3_ exhibits the highest adhesive strength of 10.7 MPa. As the content of BI is increased, the concentration of C=C radicals increases, and the crosslinking density obtained using UV irradiation also increases. Thus, the low adhesive strength of the DCA_BI_1_ is due to the low crosslinking density and the presence of unreacted components even after thermal curing [12]. By contrast, the adhesion strength of the DCA_BI_3_ is similar to, or higher than, that obtained with the same content of BA or BMA. This is because the DCA can have more cross-linking points in the BI network configuration than in the BA or BMA. Notably, the standard deviation from three separate measurements of the DCA_BI_0_, DCA_BI_1_, and DCA_BI_2_ adhesives was around 5%, decreasing to 2% for the DCA_BI_3_. This demonstrates that the bonding between the BI and the polymer network was strengthened due to the additional crosslinking [20].

### 3.5. Thermal Properties

The TGA curves of the DCA_BI_0_ and the various BI-containing DCA samples are presented in Figure 10. Here, a small initial weight loss is observed when the temperature is increased up to 300 °C, with the DCA_BI_0_ having the highest weight loss. The additional weight loss observed when the temperature is further increased to 439 °C is related to the breakdown of the C−O−C bonds and the decomposition of the copolymer carboxyl groups [21]. The initial decomposition temperatures for 5% weight loss (T_d_ 5%) and the residual weight % values at 790 °C (R_790_) are listed in Table 2. These results indicate that both the T_d_ 5% and R_790_ of the DCA_BI_3_ resin are superior to those of the DCA_BI_0_, thereby demonstrating that the thermal properties of the resin are enhanced by the inclusion of BI. However, a low content of BI does not allow the formation of a compact network, thus resulting in reduced residual weight and thermal properties.

### 3.6. Surface Properties

The pencil hardnesses of the various DCA samples are presented in Table 3. Here, the pencil hardness is seen to increase (up to 2H) with increasing BI content, which is attributed to the corresponding increase in the crosslinking density. The relatively low pencil hardness (HB) of the DCA_BI_1_ is comparable to that of the DCA_BA and DCA_BMA, and is thought to have the same underlying cause as the low adhesion (i.e., the low crosslinking density and the presence of unreacted components).

In addition, the contact angle measurements and the calculated surface energies of the PC film and the various DCA samples are presented in Table 3. During the measurement, the shape of the droplets on the surface did not change significantly, and the results indicate that the contact angle with water and ethylene glycol increases, and the surface energy decreases, with an increasing content of BI. This is because the BI molecular chain is relatively longer than those of the monomers that form the DCA network in the absence of BI [22]. Since plastics generally have low surface energies, it can be said that the decrease in surface energy due to the addition of BI in the DCA enhances the interfacial adhesion properties [23]. Moreover, the addition of BI improves the interfacial compatibility of DCA with the PC film compared to the addition of BA or BMA.

## 4. Conclusions

The successful preparation of a novel biomass itaconic acid (IA)-based epoxy acrylate oligomer (EAO), namely bisphenol A diglycidyl ether diitaconate (BI), was demonstrated, and its properties were shown to be similar or superior to those of conventional EAOs. This result provides the potential for an excellent alternative feedstock in certain plastic adhesives and coatings applications. Moreover, alternative modifications (e.g., formulation with different co-monomers) may result in further manipulation of its properties. Based on the present results, BI shows great potential as a partial replacement for petroleum-based thermosets.

## Figures and Tables

**Figure 1 polymers-15-03304-f001:**
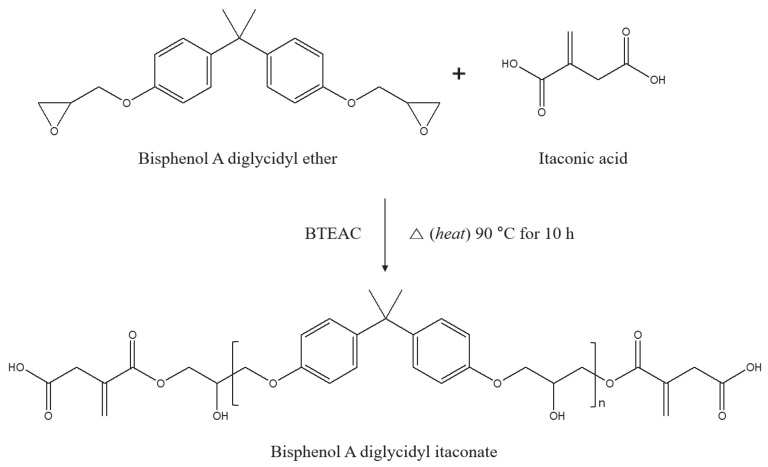
Schematic synthesis of BI from BADGE and IA.

**Figure 2 polymers-15-03304-f002:**
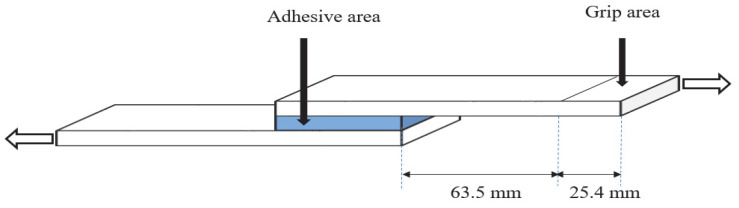
Schematic diagram of the lap-shear strength measurement.

**Figure 3 polymers-15-03304-f003:**
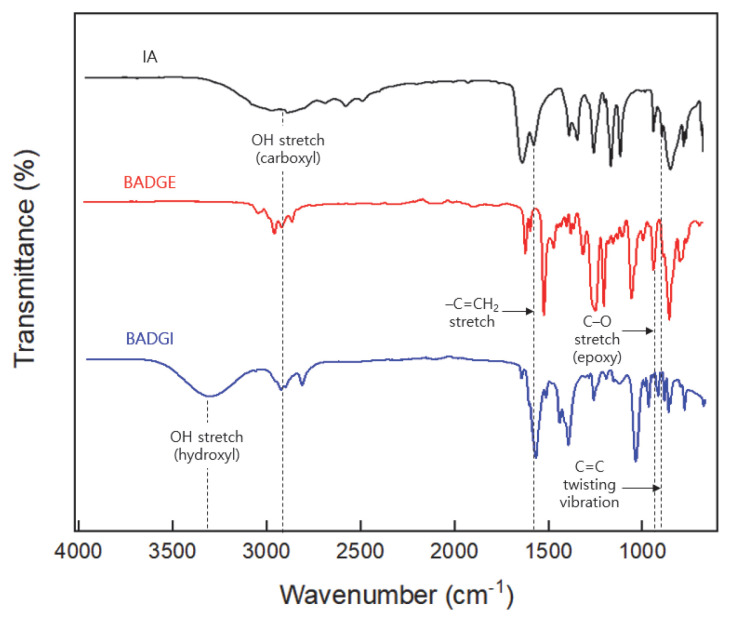
The FT-IR spectra of the IA, BADGE, and BI.

**Figure 4 polymers-15-03304-f004:**
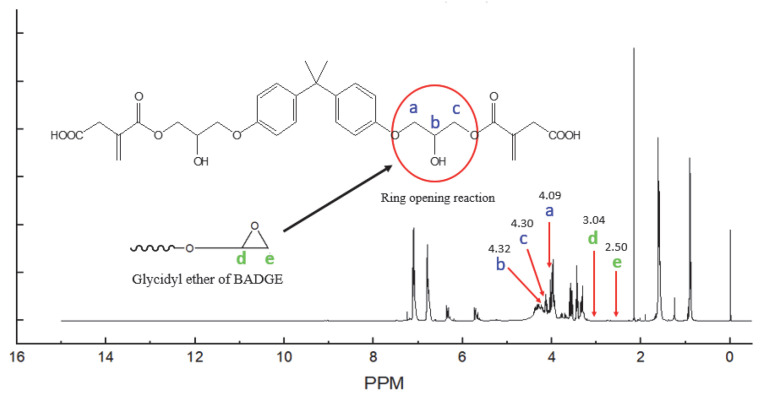
The ^1^H-NMR spectrum of the BI.

**Figure 5 polymers-15-03304-f005:**
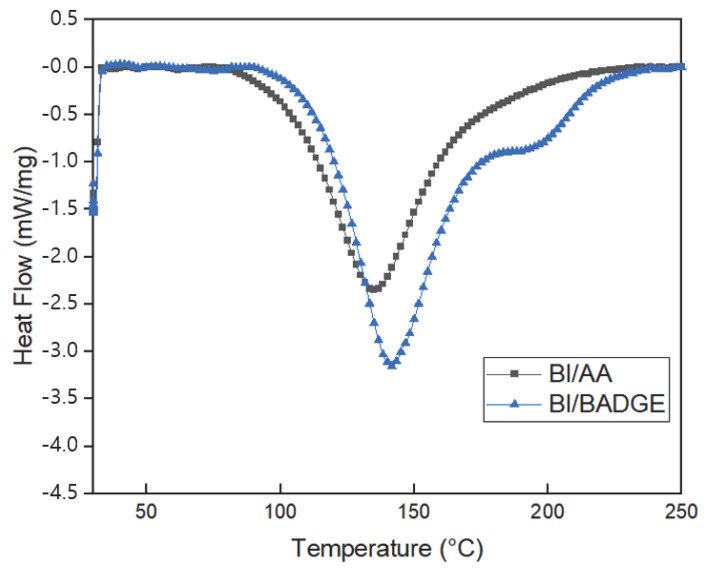
DSC thermograms showing the non-isothermal curing characteristics of the dual-cure systems in the presence and absence of BI.

**Figure 6 polymers-15-03304-f006:**
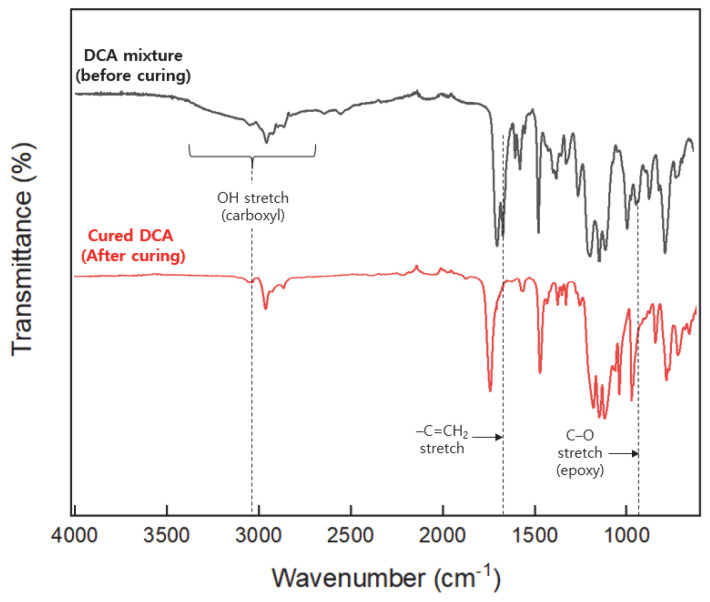
The FT-IR spectra before and after curing reation of the DCA.

**Figure 7 polymers-15-03304-f007:**
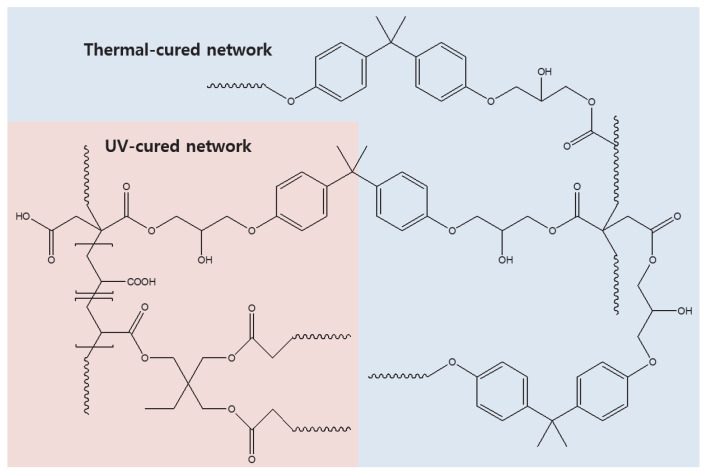
A schematic diagram of the dual-cured adhesive network with BI.

**Figure 8 polymers-15-03304-f008:**
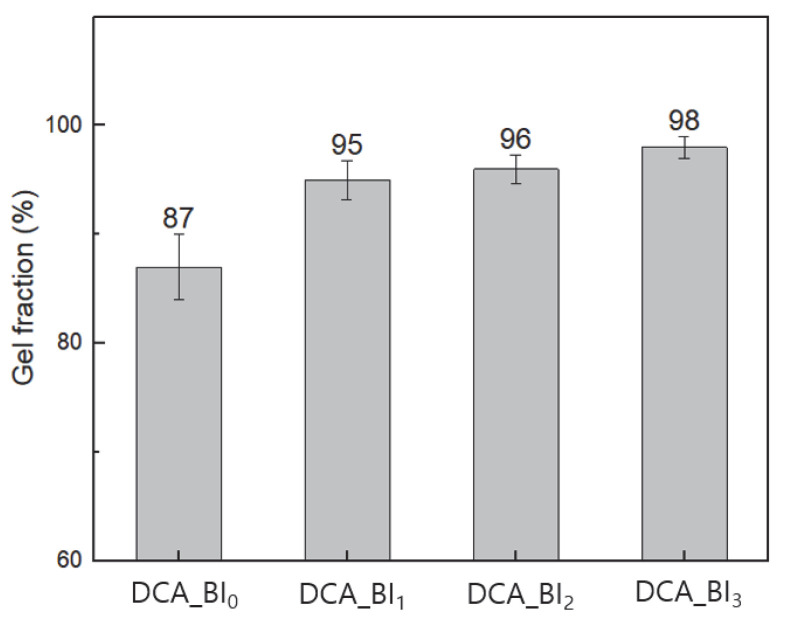
Gel fraction of various DCA samples.

**Figure 9 polymers-15-03304-f009:**
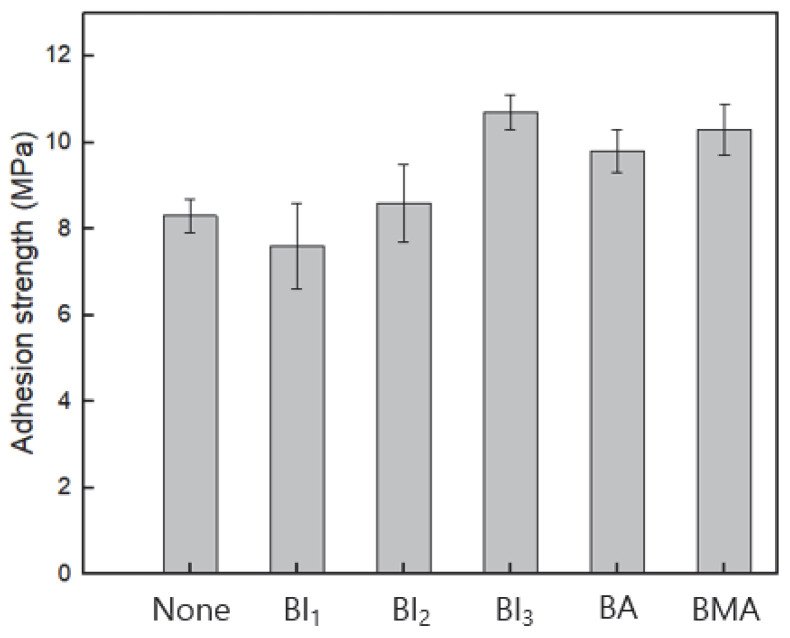
The adhesion strengths of the BA, BMA, and various DCA samples.

**Figure 10 polymers-15-03304-f010:**
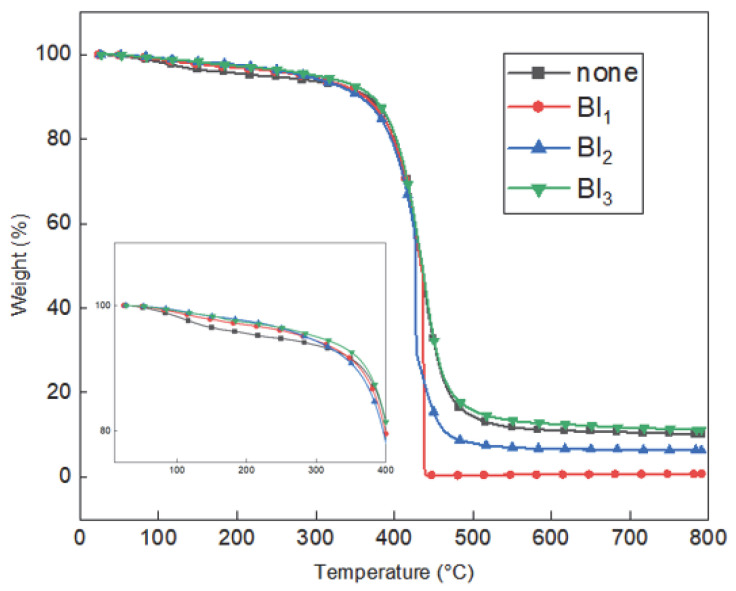
The TGA curves of the various cured DCA samples.

**Table 1 polymers-15-03304-t001:** The formulations of the various dual-cure adhesives (DCAs).

Sample	AA (mol)	TMPTA (mol)	BADGE (mol)	EAO	
				Type	Content (mol)
DCA_BI_0_	0.2	0.1	0.2	BI	0
DCA_BI_1_					0.1
DCA_BI_2_					0.2
DCA_BI_3_					0.3
DCA_BA	0.2	0.1	0.2	BA	0.3
DCA_BMA				BMA	0.3

**Table 2 polymers-15-03304-t002:** The thermal properties of the cured DCA samples.

Sample	T_d_ 5%	R_790_
DCA_BI_0_	228.2	10.2
DCA_BI_1_	283.5	0.6
DCA_BI_2_	285.5	6.4
DCA_BI_3_	300.8	11.2

**Table 3 polymers-15-03304-t003:** The pencil hardnesses and surface energies of the various DCA samples and a PC film.

Sample	Pencil Hardness	Contact Angle (°)		Surface Energy (mJ/m^2^)
		Water	Ethylene Glycol	
DCA_BI_0_	H	37.24	22.61	54.45
DCA_BI_1_	HB	42.06	26.87	52.70
DCA_BI_2_	H	53.65	34.53	48.97
DCA_BI_3_	2H	64.09	40.14	45.84
DCA_BA	2H	38.72	22.80	54.38
DCA_BMA	2H	40.12	24.22	53.81
PC film	-	79.35	56.08	35.74

## Data Availability

Not applicable.

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
