# Peer review of "Dual-Cure Adhesives Using a Newly Synthesized Itaconic Acid-Based Epoxy Acrylate Oligomer"

_polymers, 2023, doi:10.3390/polym15153304_

Round 1

Reviewer 1 Report

The authors of this paper have synthesized an adhesive of epoxy acrylate oligomer from itaconic acid which is natural and derived from biomass. They characterize the adhesive by evaluating thermal stability, pencil hardness, lap shear strength and surface energy. The findings of this paper are interesting and can be published provided the authors address my concerns:

1)      The first paragraph of the introduction needs to be improved. They start out by talking about carbon neutrality, but then talk about petroleum based epoxy. This gives very confusing views to the reader. The authors should continue on emphasizing the move towards natural products and introduce itaconic acid as a biomass-derived alternative.

2)      The authors should provide more background information about dual curing and dual cure adhesive. What are the advantages of the process, examples of DCA?

3)      What is the reaction mechanism to get the dual cure adhesive. The authors should explain that.

4)      What is the molecular weight of the resultant BI after the reaction. The authors should characterize the molecular weight distribution and comment on it.

5)      The authors attribute the difference in adhesive lap shear strength to different crosslinking density. Have the authors measured the crosslink density? They should quantify the crosslink density before making any conclusions. Same goes for surface properties.

Author Response

Response to Reviewer 1 Comments

The authors of this paper have synthesized an adhesive of epoxy acrylate oligomer from itaconic acid which is natural and derived from biomass. They characterize the adhesive by evaluating thermal stability, pencil hardness, lap shear strength and surface energy. The findings of this paper are interesting and can be published provided the authors address my concerns:

Point 1: The first paragraph of the introduction needs to be improved. They start out by talking about carbon neutrality, but then talk about petroleum based epoxy. This gives very confusing views to the reader. The authors should continue on emphasizing the move towards natural products and introduce itaconic acid as a biomass-derived alternative.

Response 1: We revised the introduction part with your advice. (p. 1~2)

Point 2: The authors should provide more background information about dual curing and dual cure adhesive. What are the advantages of the process, examples of DCA?

Response 2: With the above question, we revised the introduction part (Blue letters)

Point 3: What is the reaction mechanism to get the dual cure adhesive. The authors should explain that.

Response 3: We explained the reaction mechanism. (p. 7)

Point 4: What is the molecular weight of the resultant BI after the reaction. The authors should characterize the molecular weight distribution and comment on it.

Response 4: We characterized the molecular weight by GPC. (p. 5)

Point 5: The authors attribute the difference in adhesive lap shear strength to different crosslinking density. Have the authors measured the crosslink density? They should quantify the crosslink density before making any conclusions. Same goes for surface properties.

Response 5: We added a gel-friction graph and explanation. A convenient method for measuring insoluble fractions, such as those of bridging or network polymers. (p. 4 method, p.8 discussion)

Reviewer 2 Report

In this paper, a novel biomass-derived itaconic acid based epoxy acrylate oligomer was synthesized. Further, a dual-cure adhesive system was formulated using BADGE, AA and TMPTA with various BI contents, and the adhesive performance was investigated by measuring the thermal stability, adhesive properties, pencil hardness, and surface energy properties. The following suggestions are raised:

1. In Figure 5, the ordinate unit is power (mW). In order to eliminate the influence of sample quality and facilitate comparison, it should be normalized and divided by the initial sample quality.

2. In Figure 6, a schematic diagram of the suggested network configuration is given. To be more convincing, it is suggested to supply experiments for auxiliary verification, such as FT-IR, 1H-NMR tests at the beginning, before and after thermal curing.

3. In part 2.8. “Pencil hardness”, it is suggested to supply the reference standard for pencil hardness test.

          Minor editing of English language is required.

Author Response

Response to Reviewer 2 Comments

In this paper, a novel biomass-derived itaconic acid based epoxy acrylate oligomer was synthesized. Further, a dual-cure adhesive system was formulated using BADGE, AA and TMPTA with various BI contents, and the adhesive performance was investigated by measuring the thermal stability, adhesive properties, pencil hardness, and surface energy properties. The following suggestions are raised:

Point 1: In Figure 5, the ordinate unit is power (mW). In order to eliminate the influence of sample quality and facilitate comparison, it should be normalized and divided by the initial sample quality.

Response 1: We revised Figure 5.

Point 2: In Figure 6, a schematic diagram of the suggested network configuration is given. To be more convincing, it is suggested to supply experiments for auxiliary verification, such as FT-IR, 1H-NMR tests at the beginning, before and after thermal curing.

Response 2: We supplied new FT-IR figure and paragraphs.

Point 3: In part 2.8. “Pencil hardness”, it is suggested to supply the reference standard for pencil hardness test.

Response 3: We supplied the reference standard in the part 2.8. It is ASTM D3363 test method.
